# Vitamin Intake and Loss of Muscle Mass in Older People with Type 2 Diabetes: A Prospective Study of the KAMOGAWA-DM Cohort

**DOI:** 10.3390/nu13072335

**Published:** 2021-07-08

**Authors:** Fuyuko Takahashi, Yoshitaka Hashimoto, Ayumi Kaji, Ryosuke Sakai, Yuka Kawate, Takuro Okamura, Yuriko Kondo, Takuya Fukuda, Noriyuki Kitagawa, Hiroshi Okada, Naoko Nakanishi, Saori Majima, Takafumi Senmaru, Emi Ushigome, Masahide Hamaguchi, Mai Asano, Masahiro Yamazaki, Michiaki Fukui

**Affiliations:** 1Department of Endocrinology and Metabolism, Graduate School of Medical Science, Kyoto Prefectural University of Medicine, Kyoto 602-8566, Japan; fuyuko-t@koto.kpu-m.ac.jp (F.T.); kaji-a@koto.kpu-m.ac.jp (A.K.); sakaryo@koto.kpu-m.ac.jp (R.S.); yukawate@koto.kpu-m.ac.jp (Y.K.); d04sm012@koto.kpu-m.ac.jp (T.O.); yuri-k@koto.kpu-m.ac.jp (Y.K.); takuya07@koto.kpu-m.ac.jp (T.F.); nori-kgw@koto.kpu-m.ac.jp (N.K.); conti@koto.kpu-m.ac.jp (H.O.); naoko-n@koto.kpu-m.ac.jp (N.N.); saori-m@koto.kpu-m.ac.jp (S.M.); semmarut@koto.kpu-m.ac.jp (T.S.); emis@koto.kpu-m.ac.jp (E.U.); mhama@koto.kpu-m.ac.jp (M.H.); maias@koto.kpu-m.ac.jp (M.A.); masahiro@koto.kpu-m.ac.jp (M.Y.); michiaki@koto.kpu-m.ac.jp (M.F.); 2Department of Diabetology, Kameoka Municipal Hospital, Kyoto 621-8585, Japan; 3Department of Diabetes and Endocrinology, Matsushita Memorial Hospital, Osaka 570-8540, Japan

**Keywords:** diet, nutrition, vitamin, skeletal muscle mass, sarcopenia

## Abstract

The aim of this prospective cohort study was to examine the relationships between the intakes of various vitamins and the loss of muscle mass in older people with type 2 diabetes (T2DM). The change in skeletal muscle mass index (SMI, kg/m^2^) (kg/m^2^/year) was defined as follows: (SMI at baseline (kg/m^2^) − SMI at follow-up (kg/m^2^))/follow-up period (year). The rate of SMI reduction (%) was calculated as follows (the change in SMI (kg/m^2^/year)/SMI at baseline (kg/m^2^)) × 100. The rate of SMI reduction ≥ 1.2% was considered as the loss of muscle mass. Among 197 people with T2DM, 47.2% of them experienced the loss of muscle mass at the 13.7 ± 5.2 month follow-up. Vitamin B1 (0.8 ± 0.3 vs. 0.8 ± 0.3 mg/day, *p* = 0.031), vitamin B12 (11.2 ± 8.3 vs. 13.4 ± 7.5 μg/day, *p* = 0.049), and vitamin D (16.5 ± 12.2 vs. 21.6 ± 13.0 μg/day, *p* = 0.004) intakes in people with the loss of muscle mass were significantly lower than those without. Vitamin D intake was related to the loss of muscle mass after adjusting for sex, age, exercise, alcohol, smoking, body mass index, SMI, glucagon-like peptide-1 agonist, sodium glucose cotransporter-2 inhibitor, insulin, HbA1c, creatinine, energy intake, and protein intake (adjusted odds ratio 0.93, 95% confidence interval: 0.88–0.97, *p* = 0.003). This study showed that vitamin D intake was related to the loss of muscle mass in older people with T2DM. Vitamin B12 intake tended to be related to the loss of muscle mass, although vitamin A, vitamin B2, vitamin B6, vitamin C, and vitamin E intake were not related.

## 1. Introduction

The population of people with type 2 diabetes mellitus (T2DM), including older people, has been increasing continuously all over the world [1]. Diabetes has been shown to promote muscle catabolism due to decreasing insulin signaling and increasing insulin resistance [2], thereby causing a rapid loss of muscle strength and mass [3]. For people with T2DM, to maintain muscle mass is crucial since muscle is a main organ for body glucose metabolism [4]. Furthermore, older people with T2DM often develop sarcopenia [5], which is known as a risk of not only cardiovascular disease [6,7] but also mortality [8,9]. Thus, preventing the loss of muscle mass is a crucial therapeutic goal for people with T2DM, especially those who are older.

Macronutrient intake, such as adequate energy and protein intake, is recommended for maintenance of the body’s muscle mass [5,10,11,12]. On the other hand, micronutrients have also been associated with muscle mass [13]. Low vitamin intake is reportedly related to the risk of frailty in older people [14]. In particular, the role of vitamin D, a fat-soluble vitamin, in sarcopenia has been increasingly reported [15,16,17]. Regulation of bone homeostasis is also the role of vitamin D [16]. An epidemiological study suggested that usage of vitamin D was effective for the prevention of decreased muscle mass [17]. Serum vitamin D level was associated with physical performance and frailty in cross-sectional studies [18,19], and coexisting low serum vitamin D level and high interleukin 6 level were associated with slow gait speed in a cross-sectional study [20]. Vitamin D was reported to suppress muscle atrophy through the suppression of forkhead box protein O1 (FOXO1) in skeletal muscle [16]. Moreover, it was reported that people with sarcopenia took lower vitamin B12 than people without [21]. However, the relationships between the intakes of vitamin D or vitamin B12 and the loss of muscle mass in older people with T2DM are unknown. Additionally, the relationships between some vitamin intakes, such as vitamin A, vitamin B1, vitamin B2, vitamin B6, vitamin C and vitamin E and the loss of muscle mass are not established, although vitamin A and vitamin B6 have the effect of activation of immune function [22,23], and vitamin C and vitamin E have the effect of antioxidants [24,25]. Low serum vitamin B6 level correlates with the proinflammatory status [26]. Additionally, low vitamin C and vitamin E intakes are associated with the presence of frailty [19], and low plasma vitamin E level is also associated with the presence of the frailty [27]. Moreover, the association between vitamin B1 or vitamin B2 and the loss of muscle mass are not established. Vitamin B1, which is an essential nutrient, plays a role in various cell functions, such as energy metabolism and the breakdown of sugars and carbon skeletons [28]. Vitamin B2 plays a role in tryptophan metabolism, iron absorption, gastrointestinal tract, brain function, mitochondrial function, and other vitamins’ metabolism [29]. Therefore, this prospective cohort study investigated the relationships between the intakes of various vitamins and the loss of muscle mass in older people with T2DM. 

## 2. Method

### 2.1. Study Design

The study design of this research was a prospective cohort study.

### 2.2. Setting

The KAMOGAWA-DM cohort study, a continuing cohort study of people with diabetes mellitus, was introduced in 2014 [30]. The KAMOGAWA-DM cohort study enrolls outpatients of Kyoto Prefectural University of Medicine Hospital (Department of Endocrinology and Metabolism, Kyoto, Japan), and Kameoka Municipal Hospital (Department of Diabetology, Kameoka, Japan).

### 2.3. Study Participants

This research included participants with T2DM who answered the brief-type self-administered diet history questionnaire (BDHQ) between January 2016 and April 2018 [31]. The follow-up times were regarded as follows: the difference between the follow-up visit date and the first visit date (year). The exclusion criteria were (1) age under 65 years, (2) missing the data of body composition at baseline, (3) follow-up duration under 6 months, (4) missing follow-up body composition data, (5) inaccurate data, and (6) extremely low or high energy intake (600 or 4000 kcal/day, respectively) because of low reliability [32].

### 2.4. Ethics

Informed consent was gained from the participants, and the medical data of the participants was compiled into a database with personal information anonymized. The present study received approval from the Research Ethics Committee of Kyoto Prefectural University of Medicine (No. RBMR-E-466-6) and was carried out in conformity with the principles of the Declaration of Helsinki. 

### 2.5. Variables

The present study collected the following data. 

Sex, age, duration of diabetes, family history of diabetes, and usage medication data, including steroids, biguanides, insulin, glucagon-like peptide-1 (GLP-1) agonist, and sodium glucose cotransporter-2 (SGLT-2) inhibitors, were collected using medical records. 

Lifestyle factors were assessed by a standardized questionnaire. Based on their responses to the questionnaire, participants were grouped into non-smoker and current smoker groups. Moreover, participants were grouped into non-exercisers and exercisers based on their performance or non-performance of any kind of sport at least once a week [31].

Laboratory parameters, including fasting plasma glucose, glycosylated hemoglobin, high-density lipoprotein cholesterol, triglycerides, and creatinine were measured in venous blood samples after overnight fasting. Glycosylated hemoglobin (HbA1c) was measured by high-performance liquid chromatography. The estimated glomerular filtration rate (eGFR; mL/min/1.73 m^2^) was estimated as follows; eGFR = 194 × age^−0.287^ × serum creatinine^−1.094^ (×0.739 for women) [33].

A multifrequency impedance body composition analyzer, InBody 720 (InBody Japan, Tokyo, Japan), was used to evaluate body composition. InBody 720 was reported to be a good correlation with dual-energy X-ray absorptiometry [34]. Body mass index (BMI, kg/m^2^), dividing body weight (BW, kg) by height squared (m^2^), and skeletal muscle mass index (SMI, kg/m^2^), dividing appendicular muscle mass (kg) by height squared (m^2^), were calculated. Ideal body weight (IBW) was determined as follows: 22 × participant height squared (m^2^) [35].

Habitual food and nutrient intake were assessed using BDHQ [32], which is a dietary record for estimating the dietary intake of 58 items of the respondent during the past month: the frequency of consuming 46 food and non-alcoholic beverages; the frequency of drinking alcoholic beverages and the amount per drink of five alcoholic beverages; daily rice consumption, the type of rice, and the daily miso soup consumption; usual dietary behavior; the respondent’s regular cooking methods. A previous report showed the validity of BDHQ [36]. Data of the intake of habitual foods and nutrients included: (energy (kcal/day); fat (g/day); protein (g/day), including vegetable and animal proteins; carbohydrate (g/day); fiber (g/day); vitamin A (µg RAE/day); vitamin B1 (mg/day); vitamin B2 (mg/day); vitamin B6 (mg/day); vitamin B12 (µg/day); vitamin C (mg/day); vitamin D (µg/day); vitamin E (mg/day); alcohol consumption (g/day); supplement usage). To consider the difference in body size of each person, energy (kcal/IBW/day), fat (g/IBW/day), and carbohydrate (g/IBW/day) intakes were calculated as energy (kcal/day), fat (g/day) and carbohydrate (g/day) intakes divided by IBW (kg), respectively. Protein (g/BW/day), vegetable protein (g/BW/day), and animal protein (g/BW/day) intakes were calculated as protein (g/day), vegetable protein (g/day), and animal protein (g/day) intakes divided by BW (kg). Carbohydrate-to-fiber intake ratio was calculated as follows: dividing carbohydrate intake by fiber intake [37]. Alcohol consumption was also assessed, and we defined people with habitual alcohol consumption if they consumed alcohol >20 g/day [31]. Moreover, data on supplement intake frequency were also obtained, and habitual supplement usage was regarded as using supplements regularly. Energy intake ≥30 kcal/IBW/day was defined as adequate energy intake [12], and protein intake ≥1.2 g/BW/day defined as adequate protein intake [10,38].

### 2.6. Definition of Loss of Muscle Mass

The change in SMI (kg/m^2^/year) was calculated as follows: (SMI at baseline (kg/m^2^) − SMI at follow-up (kg/m^2^))/follow-up period (year). The rate of SMI reduction (%) was calculated as (the change in SMI (kg/m^2^/year)/SMI at baseline (kg/m^2^)) × 100 [10]. The rate of SMI reduction ≥1.2% was considered as loss of muscle mass [39].

### 2.7. Statistical Method

Categorical variables are presented as numbers and continuous variables are presented as means ± standard deviation (SD). The statistical significance of differences between groups was assessed using the chi-square test for categorical variables and student’s *t*-test for continuous variables. Evaluation of correlation was carried out with Pearson’s correlation coefficient. Logistic regression analysis was carried out to calculate the odds ratio (OR) and 95% confidence interval (CI) for the intake effect of each vitamin on the loss of muscle mass, adjusting for sex, age, habitual alcohol consumption, smoking habit, exercise habit, BMI, SMI, usage of an SGLT2 inhibitor [40], usage of a GLP-1 agonist [41], insulin treatment, HbA1c, creatinine, energy intake (kcal/IBW/day), and protein intake (g/BW/day).

Moreover, we examined the effects of vitamin intake on the loss of muscle mass according to following subgroups: exercise habit; sex; adequate energy intake (≥30 kcal/IBW/day) [12]; adequate protein intake (≥1.2 g/BW/day) [10,38]; usage of supplements; usage of biguanides. Logistic regression analyses were carried out to calculate ORs and 95% CIs adjusted for sex, age, BMI, energy intake, and protein intake.

Furthermore, the rate of SMI reduction ≥0.5% [11] and 2.0% [42] were used for other cut-off levels of loss of muscle mass.

EZR (Saitama Medical Center, Jichi Medical University, Saitama, Japan) [43] was used for statistical analyses and GraphPad Prism, version 8.4.2 (GraphPad Software, Inc., La Jolla, CA, USA), for figure creation. Differences with *p* values < 0.05 were considered statistically significant.

## 3. Results

This research enrolled 362 individuals with T2DM. We excluded 165 people: eight who did not undergo the multifrequency impedance body composition analyzer test, six who had hyper- or hypo-nutrition, fifty-five who were not followed up, one with unreliable data, two with a follow-up period under 6 months, and ninety-three who were aged under 65 years; therefore, the final study population included 197 people (112 men and 85 women) (Figure 1).

The clinical characteristics at baseline of the study participants are summarized in Table 1. Of the participants, 47.2% (*n* = 93/197) developed the loss of muscle mass at the 13.7 ± 5.2 month follow-up.

The results of habitual dietary intake assessment at baseline are shown in Table 2. Participants with the loss of muscle mass had significantly lower vitamin B1 intake (0.8 ± 0.3 vs. 0.8 ± 0.3 mg/day, *p* = 0.031), vitamin D intake (16.5 ± 12.2 vs. 21.6 ± 13.0 μg/day, *p* = 0.004), and vitamin B12 intake (11.2 ± 8.3 vs. 13.4 ± 7.5 μg/day, *p* = 0.049) than participants without the loss of muscle mass.

Logistic regression analyses showed that vitamin D intake (μg/day) was related to the loss of muscle mass (OR 0.93 (95% CI: 0.88–0.97), *p* = 0.003) after adjusting for covariates. Moreover, vitamin B1 intake (mg/day) was related to the loss of muscle mass (Model 1: OR 0.36 (95% CI: 0.14–0.93), *p* = 0.034), although it was not statistically significant after further adjusting for covariates (Model 3: OR 0.13 (95% CI: 0.01–1.23), *p* = 0.075). Vitamin B12 intake (μg/day) was related to the loss of muscle mass (Model 2: OR 0.92 (95% CI: 0.84–1.00), *p* = 0.038), although it was not statistically significant after further adjusting for covariates (Model 3: OR 0.93 (95% CI: 0.85–1.01), *p* = 0.090) (Table 3).

Moreover, we examined the effect of vitamin intake on the loss of muscle mass according to exercise habit, sex, smoking status, energy intake (≥30 kcal/IBW/day or not), protein intake (≥1.2 g/BW/day or not), usage of supplements, and usage of biguanides (Appendix A and Figure 2). Vitamin D intake (μg/day) was related to the loss of muscle mass in people both with (adjusted OR 0.92 (95% CI: 0.86–0.98), *p* = 0.014) and without exercise habits (adjusted OR 0.92 (95% CI: 0.85–0.99), *p* = 0.029). Vitamin D intake (μg/day) was related to the loss of muscle mass in women (adjusted OR 0.87 (95% CI: 0.79–0.95), *p* = 0.003) and tended to be related to the loss of muscle mass in men (adjusted OR 0.95 (95% CI: 0.89–1.01), *p* = 0.099), although the latter was not statistically significant. Whereas vitamin D intake (μg/day) was related to the loss of muscle mass in people without smoking habits (adjusted OR 0.41 ((95% CI: 0.20–0.84), *p* = 0.014), it was not related to the loss of muscle mass in people with smoking habits (adjusted OR 1.28 (95% CI: 0.03–50.23), *p* = 0.897). In addition, vitamin D intake (μg/day) was related to the loss of muscle mass in people without adequate energy intake (adjusted OR 0.89 (95% CI: 0.83–0.97), *p* = 0.005), whereas this relationship did not present in people with adequate energy intake (adjusted OR 0.95 (95% CI: 0.90–1.02), *p* = 0.137). Vitamin D intake (μg/day) was related to the loss of muscle mass in people both with (adjusted OR 0.93 (95% CI: 0.88–0.99), *p* = 0.017) and without adequate protein intake (adjusted OR 0.88 (95% CI: 0.79–0.98), *p* = 0.020). Vitamin D intake (μg/day) was related to the loss of muscle mass in people both with (adjusted OR 0.88 (95% CI: 0.78–0.99), *p* = 0.032) and without usage of supplements (adjusted OR 0.93 (95% CI: 0.87–0.98), *p* = 0.011).

Furthermore, vitamin B12 intake (μg/day) was related to the loss of muscle mass in people without adequate energy intake (adjusted OR 0.82 (95% CI: 0.70–0.95), *p* = 0.011). Moreover, vitamin B12 intake (μg/day) was related to the loss of muscle mass in people with usage of supplements (adjusted OR 0.71 (95% CI: 0.53–0.94), *p* = 0.016), whereas this relationship did not present in people without usage of supplements (adjusted OR 0.96 (95% CI: 0.88–1.05), *p* = 0.368). Vitamin B12 intake (μg/day) was related to the loss of muscle mass in older people without usage of biguanides (adjusted OR 0.88 (95% CI: 0.79–0.97), *p* = 0.013), whereas this relationship did not present in older people with usage of biguanides (adjusted OR 1.04 (95% CI: 0.90–1.21), *p* = 0.569) (Appendix A).

All analyses were adjusted for sex, age, body mass index, energy intake, and protein intake.

Moreover, Table 4 shows the results of using other cut-off levels of the loss of muscle mass of the rate of SMI reduction, ≥0.5% and ≥2.0%. Vitamin D intake (μg/day) was related to the loss of muscle mass even at different cut-off levels (cut-off point, rate of SMI reduction ≥ 0.5%, adjusted OR 0.94 (95% CI 0.89–0.99), *p* = 0.020; cut-off point, rate of SMI reduction ≥ 2.0%, adjusted OR 0.93 (95% CI 0.88–0.98), *p* = 0.006).

The correlation between vitamin D intake and the rate of SMI reduction are investigated in Figure 3. Vitamin D intake was correlated with the rate of SMI reduction (*r* = −0.242, *p* < 0.001).

## 4. Discussion

The present study investigated the relationship between the intake of each vitamin and the loss of muscle mass and demonstrated that participants with experience of the loss of muscle mass had lower vitamin D intake than those without, and lower vitamin D intake was related to the loss of muscle mass even after adjusting for covariates, including energy and protein intakes. In particular, older people with T2DM have a higher risk of the incidence of the loss of muscle mass and sarcopenia. However, adequate vitamin D intake may avoid the incidence of the loss of muscle mass. 

Diabetes causes loss of muscle strength and mass [3], promoting muscle catabolism due to decreasing insulin signaling and increasing insulin resistance [2]. However, muscle is a primary organ for glucose metabolism [4], and people with T2DM should maintain muscle mass. Vitamin D is important not only for bone formation but also for the maintenance of skeletal muscle mass [15]. Vitamin D supplementation has the effect of increasing muscle mass [44]. Older people tend to have lower vitamin D levels due to low dietary intake and reduced ultraviolet irradiation of the skin [45,46]. In addition, previous studies have revealed that low serum vitamin D levels increased the risks of loss of muscle strength and mass and frailty [47,48], although serum vitamin D levels were not measured in the present study. It has been demonstrated that vitamin D receptor (VDR), which is activated in the presence of 1,25-dihydroxyvitamin D, the active form of vitamin D [16], is present in human skeletal muscle [49]. A recent finding showed that VDR-knockout mice showed insulin resistance with an increased FOXO1 expression in skeletal muscle [50]. Moreover, previous studies have revealed that supplementation of vitamin D in people with prediabetes or insulin resistance and inadequate vitamin D levels improves insulin sensitivity [51,52,53,54,55]. Thus, vitamin D may improve insulin sensitivity in skeletal muscle and suppress the loss of muscle mass. It has been suggested that vitamin D signaling regulates gene transcription through the VDR and also activates intracellular signaling pathways related to calcium metabolism, which is related to myoblast proliferation and differentiation [56]. Therefore, adequate vitamin D intake potentially prevents loss of skeletal muscle mass and the development of sarcopenia.

Moreover, there was an association between vitamin D intake and the loss of muscle mass in older people without adequate energy intake, whereas this was not related to the loss of muscle mass in older people with adequate energy intake. In addition to protein intake, previous reports have revealed that energy intake is also related to muscle mass reduction [3,5]. Energy intake reduction prevents the Akt-dependent signaling pathway, leading to the impairment of protein synthesis in the muscle and to the promotion of collagen accumulation and skeletal muscle fibrosis [57,58]. Thus, there is a possibility that vitamin D intake may suppress the loss of muscle mass in older people without adequate energy intake. Vitamin D was related to the loss of muscle mass in both older people with usage of supplements and those without in this study. Therefore, vitamin D may be effective in preventing the loss of muscle mass regardless of supplement usage. In this study, vitamin D intake was related to the loss of muscle mass in non-smokers only. Cigarette smoke extracts have the ability to inhibit vitamin D-induced VDR translocation and impair several vitamin D effects on skeletal muscle cells [59,60]. This suggests that VDR on skeletal muscle is inhibited in smokers, thus preventing vitamin D intake from influencing the suppression of the loss of muscle mass. 

In this study, vitamin B12 intake was related to the loss of muscle mass, although the results of multivariate analysis were not statistically significant. In addition, vitamin B12 was related to the loss of muscle mass in older people without adequate energy intake. Vitamin B12 is known to affect sarcopenia, because it helps to lower serum levels of homocysteine, higher levels of which are related to the loss of muscle strength and gait speed [61,62,63]. Although homocysteine is not included in the diet, it is an essential intermediate in normal mammalian metabolism of methionine [64]. Homocysteine also has a negative effect on cardiovascular disease through its adverse effects on cardiovascular endothelium and smooth muscle cells [65]. The use of metformin for a long time is a risk factor for vitamin B12 deficiency [66]. In this study, vitamin B12 was related to the loss of muscle mass in older people without usage of biguanides, whereas this association was absent in older people with usage of biguanides. The effect of vitamin B12 on inhibiting the loss of muscle mass may be lower in older people with usage of biguanides. Therefore, vitamin B12 may be related to the loss of muscle mass; nevertheless, further study is warranted. 

In the present study, vitamin B1 intake was related to the loss of muscle mass, although the results of multivariate analysis were not statistically significant. Insufficient vitamin B1 causes inadequate glucose metabolism in mitochondria [28] and low mitochondrial function leads to muscle loss [67]. However, further research about the relationship between vitamin B1 intake and the loss of muscle mass is needed. 

In this study, vitamin A, vitamin C, and vitamin E intakes were not associated with the loss of muscle mass. Plasma vitamin C level is low in people with diabetes mellitus because of impaired hepatic and renal regeneration, as well as increased urinary excretion and impaired hepatic biosynthesis of vitamin C [68]. Thus, vitamin C intake might not be associated with the loss of muscle mass in people with T2DM. Additionally, the effect of vitamin intake for the loss of muscle mass may be smaller than for people in general, because T2DM is associated with oxidative stress [69]. On the other hand, vitamin A, vitamin E, and vitamin C intake were associated with frailty in a previous study [14]; nevertheless, further study is needed.

This research has certain limitations. First, this study is an observational study. Consequently, the follow-up period was different for each participant, and we adjusted the differences in the follow-up period by using the rate of SMI reduction per year. Furthermore, there might have been unknown confounding factors. Second, the data on eating habits used in this study were obtained at baseline; thus, there is a possibility that eating habits changed during the follow-up period. Furthermore, with the data on eating habits being self-reported, eating habits might not be evaluated exactly. However, a previous report showed the specifications and validity of BDHQ [36]. Third, the supplements used by participants were not identified. Finally, the use of questionnaires without any objective blood measures might also be a risk of bias since we did not evaluate any objective blood measures, such as serum vitamin D, vitamin B12, and homocysteine levels. Additionally, the present study did not consider vitamin D produced in the skin by solar energies under the ultraviolet ray’s action. Therefore, the present study could not account for total vitamin D status in the body.

## 5. Conclusions

The present study identified that vitamin D intake is associated with the loss of muscle mass in older people with T2DM; and so, there is a possibility that this result might not apply to a general population. To prevent the loss of muscle mass in older people with T2DM, having a higher risk of the loss of muscle mass, we should consider not only macronutrient intake, such as energy and protein intakes, but also micronutrient intake, such as vitamin D intake in older people with T2DM.

## Figures and Tables

**Figure 1 nutrients-13-02335-f001:**
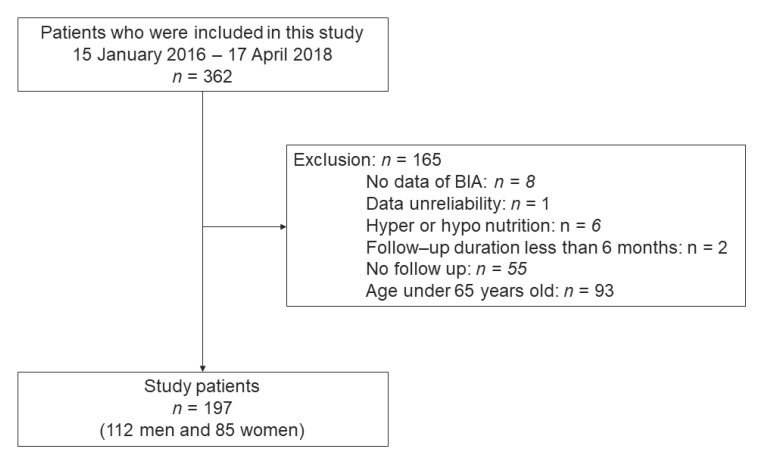
Inclusion and exclusion flow.

**Figure 2 nutrients-13-02335-f002:**
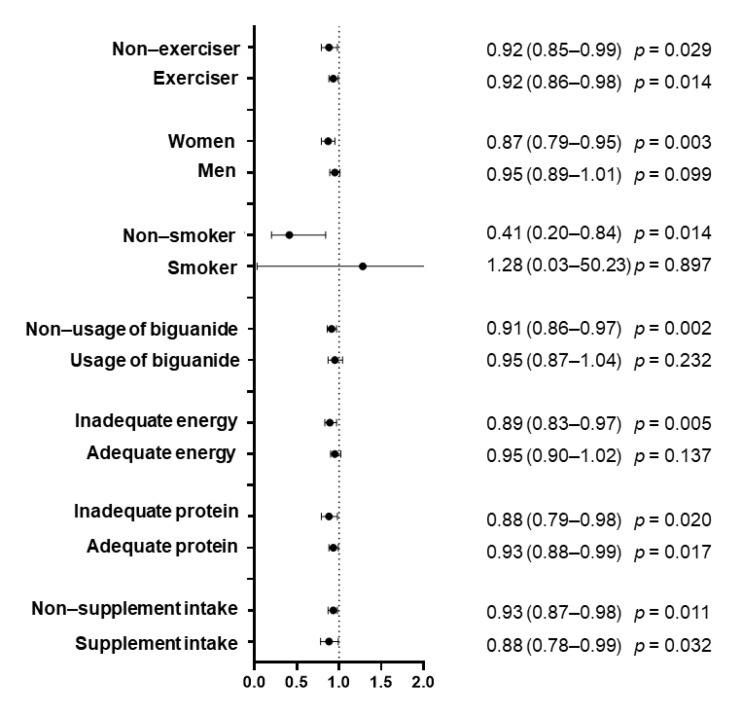
Odds ratio of vitamin D intake on the loss of muscle mass according to the groups.

**Figure 3 nutrients-13-02335-f003:**
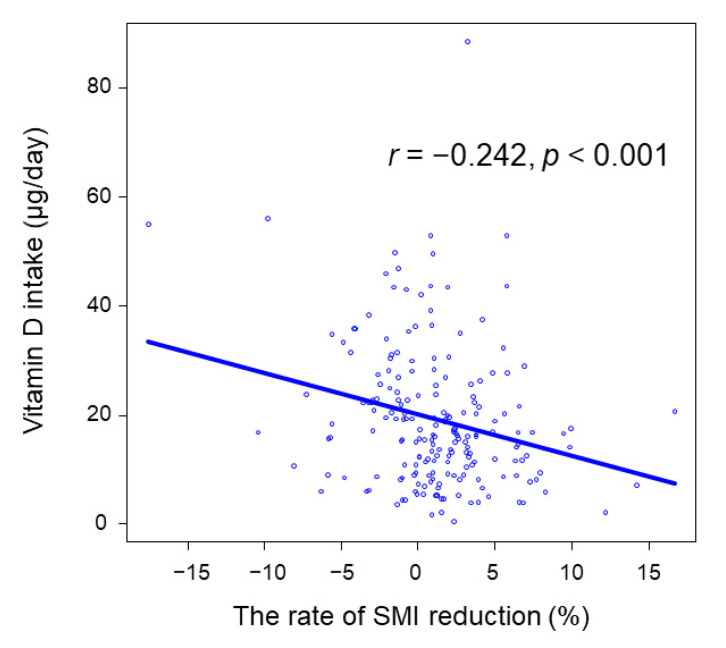
The correlation between vitamin D intake and the rate of SMI reduction. SMI, skeletal muscle mass index.

**Table 1 nutrients-13-02335-t001:** Clinical characteristics at baseline of study participants.

	All *n*= 197	Loss of Muscle Mass (−) *n* = 104	Loss of Muscle Mass (+) *n* = 93	*p*
Age (years)	72.3 (5.2)	72.3 (5.1)	72.3 (5.3)	0.983
Sex (men/women)	112/85	54/50	58/35	0.182
Duration of diabetes (years)	16.1 (10.1)	15.5 (9.6)	16.8 (10.6)	0.381
Family history of diabetes (−/+)	115/82	62/42	53/40	0.819
BMI (kg/m^2^)	23.7 (3.9)	23.6 (3.2)	23.9 (4.5)	0.529
Appendicular muscle mass at baseline (kg)	19.6 (4.3)	17.3 (4.0)	18.2 (3.6)	0.103
Appendicular muscle mass at follow-up (kg)	17.5 (3.9)	17.6 (4.1)	17.4 (3.7)	0.700
SMI at baseline (kg/m^2^)	6.9 (0.9)	6.7 (1.0)	7.0 (0.9)	0.048
SMI at follow-up (kg/m^2^)	6.8 (1.0)	6.8 (1.0)	6.7 (0.9)	0.212
SBP (mmHg)	134.0 (18.9)	133.8 (19.1)	134.3 (18.7)	0.873
DBP (mmHg)	77.1 (10.5)	77.0 (11.2)	77.3 (9.7)	0.830
Insulin (−/+)	144/53	77/27	67/26	0.877
Sodium glucose cotransporter-2 inhibitor (−/+)	172/25	91/13	81/12	1.000
Glucagon-like peptide-1 agonist (−/+)	174/23	88/16	86/7	0.136
Biguanides (−/+)	132/65	65/39	67/26	0.204
Steroids usage (−/+)	192/5	103/1	89/4	0.301
Habit of smoking (−/+)	169/28	90/14	79/14	0.908
Habit of exercise (−/+)	85/112	41/63	44/49	0.331
Triglycerides (mmol/L)	1.4 (0.9)	1.4 (0.9)	1.5 (1.0)	0.755
HDL cholesterol (mmol/L)	1.6 (0.4)	1.6 (0.4)	1.6 (0.4)	0.820
Blood glucose levels (mmol/L)	8.2 (2.8)	8.2 (2.6)	8.2 (3.1)	0.866
HbA1c (mmol/mol)	55.3 (10.7)	54.9 (10.7)	55.8 (10.7)	0.533
HbA1c (%)	7.2 (1.0)	7.2 (1.0)	7.3 (1.0)	0.533
Creatinine (umol/L)	74.8 (26.1)	71.7 (23.9)	78.4 (28.0)	0.070
eGFR (mL/min/1.73 m^2^)	65.6 (17.5)	67.0 (17.0)	64.1 (18.0)	0.241
Uric acid (mmol/L)	310.4 (71.3)	307.2 (66.7)	313.0 (76.2)	0.513

Data were expressed as number or mean (standard deviation). The difference between groups was assessed by chi-square test or Student’s *t*-test. BMI, Body mass index; SMI, Skeletal muscle mass index; SBP, systolic blood pressure; DBP, diastolic blood pressure; HDL, high-density lipoprotein; eGFR, estimated glomerular filtration rate.

**Table 2 nutrients-13-02335-t002:** Habitual dietary intake at baseline of study participants.

	All *n* = 197	Loss of Muscle Mass (−) *n* = 81	Loss of Muscle Mass (+) *n* = 93	*p*
Total energy (kcal/day)	1746.4 (596.3)	1818.8 (596.1)	1665.4 (589.2)	0.071
Energy (kcal/IBW/day)	30.9 (10.0)	32.5 (10.1)	29.2 (9.7)	0.021
Total protein (g/day)	75.3 (30.3)	79.5 (29.0)	70.6 (31.3)	0.040
Protein (g/BW/day)	1.3 (0.6)	1.4 (0.5)	1.2 (0.6)	0.079
Protein per energy (%)	17.3 (3.6)	17.8 (4.1)	16.8 (3.0)	0.058
Animal protein (g/day)	47.2 (23.8)	50.7 (23.1)	43.2 (24.1)	0.026
Animal protein (g/BW/day)	0.8 (0.4)	0.9 (0.4)	0.7 (0.5)	0.045
Vegetable protein (g/day)	28.1 (9.7)	28.7 (9.7)	27.4 (9.8)	0.348
Vegetable protein (g/BW/day)	0.5 (0.2)	0.5 (0.2)	0.5 (0.2)	0.285
Animal/vegetable protein ratio	1.8 (1.0)	1.9 (1.2)	1.6 (0.6)	0.017
Total fat (g/day)	55.6 (22.1)	58.2 (21.0)	52.8 (23.1)	0.088
Fat (g/IBW/day)	1.0 (0.4)	1.1 (0.4)	0.9 (0.4)	0.034
Fat per energy (%)	28.9 (6.5)	29.3 (6.8)	28.4 (6.1)	0.300
Total carbohydrate (g/day)	218.1 (82.2)	225.3 (86.0)	210.1 (77.3)	0.197
Carbohydrate (g/IBW/day)	3.9 (1.4)	4.0 (1.5)	3.7 (1.2)	0.089
Carbohydrate per energy (%)	50.2 (9.1)	49.3 (9.7)	51.2 (8.3)	0.138
Dietary fiber (g/day)	12.5 (5.3)	12.9 (5.4)	12.1 (5.3)	0.308
Carbohydrate/fiber ratio	18.9 (7.1)	18.8 (6.8)	19.1 (7.4)	0.752
Alcohol consumption (g/day)	7.5 (17.4)	8.1 (19.3)	6.9 (15.1)	0.631
Vitamin A (µg RAE/day)	807.6 (612.0)	824.7 (596.1)	788.4 (632.0)	0.679
Vitamin E (mg/day)	7.9 (3.1)	8.2 (3.0)	7.6 (3.2)	0.130
Vitamin B1 (mg/day)	0.8 (0.3)	0.8 (0.3)	0.8 (0.3)	0.031
Vitamin B2 (mg/day)	1.5 (0.6)	1.5 (0.5)	1.4 (0.6)	0.175
Vitamin B6 (mg/day)	1.4 (0.6)	1.4 (0.5)	1.3 (0.6)	0.075
Vitamin B12 (µg/day)	12.4 (8.0)	13.4 (7.5)	11.2 (8.3)	0.049
Vitamin C (mg/day)	126.2 (62.5)	129.7 (66.2)	122.4 (58.3)	0.412
Vitamin D (µg/day)	19.2 (12.9)	21.6 (13.0)	16.5 (12.2)	0.004

Data were expressed as number or mean (standard deviation). The difference between groups was assessed by chi-square test or Student’s *t* test. IBW, ideal body weight; BW, body weight.

**Table 3 nutrients-13-02335-t003:** Odds ratio of the quantity of vitamin intake on the presence of the loss of muscle mass.

	Model 1		Model 2		Model 3	
	OR (95% CI)	*p*	OR (95% CI)	*p*	OR (95% CI)	*p*
Vitamin A (µg RAE/day)	1.00 (1.00–1.00)	0.679	1.00 (1.00–1.00)	0.637	1.00 (1.00–1.00)	0.471
Vitamin E (mg/day)	0.93 (0.85–1.02)	0.132	0.96 (0.80–1.14)	0.616	0.95 (0.79–1.13)	0.556
Vitamin B1 (mg/day)	0.36 (0.14–0.93)	0.034	0.15 (0.02–1.28)	0.083	0.13 (0.01–1.23)	0.075
Vitamin B2 (mg/day)	0.70 (0.42–1.17)	0.176	0.79 (0.25–2.51)	0.694	0.77 (0.23–2.53)	0.664
Vitamin B6 (mg/day)	0.63 (0.38–1.05)	0.078	0.42 (0.11–1.57)	0.197	0.43 (0.11–1.72)	0.233
Vitamin B12 (µg/day)	0.96 (0.93–1.00)	0.054	0.92 (0.84–1.00)	0.038	0.93 (0.85–1.01)	0.090
Vitamin C (mg/day)	1.00 (0.99–1.00)	0.411	1.00 (1.00–1.01)	0.996	1.00 (0.99–1.01)	0.953
Vitamin D (µg/day)	0.97 (0.94–0.99)	0.006	0.91 (0.87–0.96)	<0.001	0.93 (0.88–0.97)	0.003

Model 1 is unadjusted; Model 2 is adjusted for sex, age, insulin treatment, body mass index, skeletal muscle mass index, smoking habit, exercise habit, energy intake, protein intake, habitual alcohol consumption; Model 3 is adjusted for covariates of Model 2 and creatinine, HbA1c, GLP-1 agonist, SGLT2 inhibitor.

**Table 4 nutrients-13-02335-t004:** Odds ratio of the quantity of vitamin intake on the presence of the loss of muscle mass according to the difference cut-off point.

**Cut-Off Point; Rate of SMI Reduction ≥ 0.5%**						
	**Model 1**		**Model 2**		**Model 3**	
	**OR (95% CI)**	***p***	**OR (95% CI)**	***p***	**OR (95% CI)**	***p***
Vitamin A (µgRAE/day)	1.00 (1.00–1.00)	0.770	1.00 (1.00–1.00)	0.667	1.00 (1.00–1.00)	0.684
Vitamin E (mg/day)	0.94 (0.85–1.03)	0.166	0.94 (0.79–1.13)	0.517	0.94 (0.78–1.14)	0.535
Vitamin B1 (mg/day)	0.48 (0.19–1.20)	0.117	0.29 (0.04–2.33)	0.243	0.32 (0.04–2.79)	0.303
Vitamin B2 (mg/day)	0.72 (0.43–1.21)	0.214	0.53 (0.16–1.69)	0.283	0.56 (0.17–1.91)	0.355
Vitamin B6 (mg/day)	0.71 (0.43–1.16)	0.170	0.57 (0.16–2.09)	0.399	0.68 (0.18–2.59)	0.570
Vitamin B12 (µg/day)	0.97 (0.93–1.01)	0.090	0.93 (0.86–1.01)	0.101	0.94 (0.86–1.02)	0.145
Vitamin C (mg/day)	1.00 (1.00–1.00)	0.901	1.00 (1.00–1.01)	0.388	1.00 (1.00–1.01)	0.327
Vitamin D (µg/day)	0.97 (0.95–1.00)	0.021	0.94 (0.89–0.98)	0.007	0.94 (0.89–0.99)	0.020
**Cut-Off Point; Rate of SMI Reduction ≥ 2.0%**						
	**Model 1**		**Model 2**		**Model 3**	
	**OR (95% CI)**	***p***	**OR (95% CI)**	***p***	**OR (95% CI)**	***p***
Vitamin A (µgRAE/day)	1.00 (1.00–1.00	0.563	1.00 (1.00–1.00)	0.883	1.00 (1.00–1.00)	0.727
Vitamin E (mg/day)	0.95 (0.87–1.05)	0.335	0.96 (0.80–1.14)	0.635	0.94 (0.78–1.12)	0.485
Vitamin B1 (mg/day)	0.56 (0.21–1.47)	0.237	0.40 (0.05–3.47)	0.406	0.32 (0.03–3.00)	0.320
Vitamin B2 (mg/day)	0.75 (0.44–1.28)	0.291	0.60 (0.18–1.98)	0.399	0.61 (0.18–2.10)	0.435
Vitamin B6 (mg/day)	0.83 (0.50–1.38)	0.474	1.05 (0.28–3.95)	0.946	1.11 (0.28–4.37)	0.887
Vitamin B12 (µg/day)	0.97 (0.94–1.01)	0.188	0.92 (0.85–1.00)	0.063	0.93 (0.85–1.02)	0.120
Vitamin C (mg/day)	1.00 (1.00–1.00)	0.927	1.00 (1.00–1.01)	0.703	1.00 (0.99–1.01)	0.869
Vitamin D (µg/day)	0.98 (0.95–1.00)	0.054	0.92 (0.88–0.97)	0.002	0.93 (0.88–0.98)	0.006

Model 1 is unadjusted; Model 2 is adjusted for sex, age, body mass index, skeletal muscle mass index, habit of smoking, habit of exercise, insulin treatment, energy intake, protein intake, habitual alcohol consumption; Model 3 is adjusted for covariates of Model 2 and creatinine, HbA1c, GLP-1 agonist, SGLT2 inhibitor.

## Data Availability

The datasets generated during and/or analyzed during the current study are available from the corresponding author on reasonable request.

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
