# Peer review of "Vitamin Intake and Loss of Muscle Mass in Older People with Type 2 Diabetes: A Prospective Study of the KAMOGAWA-DM Cohort"

_nutrients, 2021, doi:10.3390/nu13072335_

Round 1
Reviewer 1 Report
People with diabetes mellitus are more at risk for muscle mass loss. As a low muscle mass could lead to functional decline, it is crucial to prevent it. The relationship between micronutrients intake and muscle mass is not clear, especially in diabetes patients. Thus, this prospective cohort study aims to investigate the relationships between the intakes of various vitamins and muscle mass loss in people over 65 years with diabetes. English writing is good and the paper is well written. However, there are major methodological problems that question the validity of the results.
Method section:
The authors defined a significant loss of muscle mass as reduction of >1.2% of skeletal mass index. What is the clinical relevance of this threshold?
After the assessment at baseline, it is not clear at what point in the follow-up the body composition took place. Indeed, if a subject has been followed for a longer period of time, he is at greater risk of losing muscle mass. Therefore, the authors should use a cox regression analysis and not a logistic regression, or fixe the same length of follow-up for all subjects (for example: body composition at 6 months of follow-up).
The authors examine the relationship between vitamins intake at baseline and muscle mass loss at “x” time of follow-up. How can we be sure that eating habits have not changed during the follow-up? This is a major limitation for this study and questions the validity of the results. The use of questionnaires without any objective blood measures is also at high risk of bias.
Author Response
Response to Reviewer 1
Point 1
People with diabetes mellitus are more at risk for muscle mass loss. As a low muscle mass could lead to functional decline, it is crucial to prevent it. The relationship between micronutrients intake and muscle mass is not clear, especially in diabetes patients. Thus, this prospective cohort study aims to investigate the relationships between the intakes of various vitamins and muscle mass loss in people over 65 years with diabetes. English writing is good and the paper is well written. However, there are major methodological problems that question the validity of the results.
Thank you for your valuable comments. We have revised the manuscripts, according to your comments as below.
Method section:
The authors defined a significant loss of muscle mass as reduction of >1.2% of skeletal mass index. What is the clinical relevance of this threshold?
Response
Thank you for your valuable comment. Previous studies showed that muscle mass decreases with age, especially in older people, by 0.5% to 2% per year. Thus, we have set the cut-off of 1.2% per year. However, as you say, the cut-off of 1.2% might be seen as arbitrary. Therefore, according to your comment, we have added other cut-off points of 0.5% and 2%, which have been shown as the cut-off of muscle mass decline in previous studies. Using these cut-off values, similar results were obtained. According to your comment, we have added these points in the Method and Results, including new Table 4, sections described as below.
Method (Line: 178-179)
“Furthermore, the rate of SMI change ≥0.5% [11] and 2.0% [35] were used for other cut-off levels of loss of muscle mass.”
Results (Line: 273-378)
“Moreover, Table 4 shows the results of using other cut-off levels of loss of muscle mass of the rate of SMI change ≥0.5% and ≥2.0%. Vitamin D intake was associated with incident loss of muscle mass even at different cutoff levels (cut off point; rate of SMI reduction ≥0.5%, adjusted OR 0.94 [95% CI 0.89-0.99], p = 0.020; and cut off point; rate of SMI reduction ≥2.0%, adjusted OR 0.93 [95% CI 0.88-0.98], p = 0.006).”
References
- von Haehling, S.; Morley, J. E.; Anker, S. D. An overview of sarcopenia: facts and numbers on prevalence and clinical impact. J Cachexia Sarcopenia Muscle. 2010, 1, 129-133.
Point 2
After the assessment at baseline, it is not clear at what point in the follow-up the body composition took place. Indeed, if a subject has been followed for a longer period of time, he is at greater risk of losing muscle mass. Therefore, the authors should use a cox regression analysis and not a logistic regression, or fixe the same length of follow-up for all subjects (for example: body composition at 6 months of follow-up).
Response
Thank you for your comment. As you say, the longer the subject is followed, the greater the risk of muscle mass loss. This study is an observational study and follow-up period was different for each participant. Therefore, we adjusted the differences in the follow-up period by using the rate of SMI reduction per year, which was calculated by dividing the reduction in SMI by the follow-up period (year). We have mentioned this point as one of the limitations of this study in the Discussion section described as below.
Discussion (Line: 367-371)
“First, the present study is an observational study. Consequently, follow-up period was different for each participant, and we adjusted the differences in the follow-up period by using the rate of SMI reduction per year. Furthermore, there might have been unknown confounding factors.”
Point 3
The authors examine the relationship between vitamins intake at baseline and muscle mass loss at “x” time of follow-up. How can we be sure that eating habits have not changed during the follow-up? This is a major limitation for this study and questions the validity of the results. The use of questionnaires without any objective blood measures is also at high risk of bias.
Response
Thank you for your comment. As you say, this study examined the relationship between vitamins intake at baseline and muscle mass loss of follow-up and no data on eating habits during and at follow-up. Unfortunately, however, in this study, the data on eating habits were obtained at baseline only, and we could not be sure that eating habits have not changed during the follow-up, as we mentioned as one of the limitations of this study in the Discussion section.
Furthermore, as you say, the use of questionnaires without any objective blood measures might also be a risk of bias. Unfortunately, however, we did not evaluate any objective blood measures, such as serum vitamin D, vitamin B12, and homocysteine levels. Thus, we have mentioned this point as one of the limitations of this study in the Discussion section described as below.
Discussion (Line: 378-381)
“Finally, the use of questionnaires without any objective blood measures might also be a risk of bias, since we did not evaluate any objective blood measures, such as serum vitamin D, vitamin B12, and homocysteine levels.”
Reviewer 2 Report
Thank you for giving me the opportunity to review this manuscript studying the association between vitamin intake and muscle mass reduction in a cohort of Japanese patients with type 2 diabetes mellitus.
GENERAL COMMENTS
The manuscript should be revised by a Native English Speaker. Without going into details, there are some grammatical and lexical errors (first line of the abstract, objectives (the use of the verb ‘to object’), and others). I strongly encourage authors to follow the STROBE recommendations to report their results (abstract and manuscript). Please, avoid to display Tables on separate pages.
ABSTRACT:
- A clear objective in both the abstract and the manuscript are mandatory.
- Although reported, the relation between low vitamin intake and the presence of sarcopenia in older people is not as clear as authors seem to state. I recommend to remove the two first sentences and start the abstract clearly informing about the objective of this study.
- Please, revise the formula in the abstract (it is not exactly the same than that of the manuscript).
- Sentence beginning with a number, this should be written with all the words (e.g. 47% experienced...). Please, include 47% of patients (individuals, participants…) when using percentages. This information may be obvious for authors, but not necessarily for the readers.
- The main study variables should be mentioned in the Methods.
- Conclusions should clearly respond to the previously defined objectives.
INTRODUCTION
- The rationale should be further developed. The authors establish an excessively simplistic relationship between vitamin intake and muscle mass reduction.
- Please, the revise the use of the verb ‘to object’ (nothing to do with an objective). Try to be more specific when stating the objective: ‘This prospective cohort study objected to investigate the relationships between the intakes of various vitamins and muscle mass loss in older people with T2DM’: what vitamins? If the objective was only this one, why do you register other nutrients?
METHODS
- I strongly encourage authors to follow the STROBE recommendations to report their results.
- The subsection ‘Study participants’ should refer only to participants (inclusion and exclusion criteria). ‘Study design’ and ‘Ethics’ should be reported separately.
- Please, make a clear distinction between the study variables and potential confounders.
- The authors divide participants into two groups according to muscle mass reduction, please state this clearly in the Methods. Why using the cutoff of 1.2% for muscle mass reduction?
- Statistical Analysis: I cannot see the difference between lines 146-152 and lines 153-158. The three models described in Table 3, should be also in the text. Remember, that Tables and manuscript should be understood separately.
RESULTS
- Figure 2. Odds ratio of vitamin D intake on incident muscle mass loss according to the groups. Which groups?
- Data included in Figure 2 deserve further explanation in the manuscript.
DISCUSSION
- The brief-type self-administered diet history questionnaire (BDHQ) seems to have important limitations. Please, if available, include metric proprieties and validation studies in Methods and include a comment in the Discussion regarding self-reported methods.
Author Response
Response to Reviewer 2
Point 1
GENERAL COMMENTS
The manuscript should be revised by a Native English Speaker. Without going into details, there are some grammatical and lexical errors (first line of the abstract, objectives (the use of the verb ‘to object’), and others). I strongly encourage authors to follow the STROBE recommendations to report their results (abstract and manuscript). Please, avoid to display Tables on separate pages.
Response
Thank you for your valuable suggestion. According to your suggestion, we have corrected grammatical and lexical errors. In addition, we have revised our manuscript according to the STROBE statement, and we have displayed all Tables on one paper.
Point 2
ABSTRACT:
A clear objective in both the abstract and the manuscript are mandatory.
Although reported, the relation between low vitamin intake and the presence of sarcopenia in older people is not as clear as authors seem to state. I recommend to remove the two first sentences and start the abstract clearly informing about the objective of this study.
Response
Thank you for your valuable suggestion. According to your suggestion, we have removed the two first sentences, and revised the objective of this study to clearly describe in the Abstract section described as below.
Abstract (Line: 18-20)
“The purpose of this prospective cohort study was to investigate the relationships between the intakes of various vitamins and muscle mass loss in older people with type 2 diabetes (T2DM).”
Point 3
Please, revise the formula in the abstract (it is not exactly the same than that of the manuscript).
Sentence beginning with a number, this should be written with all the words (e.g. 47% experienced...). Please, include 47% of patients (individuals, participants…) when using percentages. This information may be obvious for authors, but not necessarily for the readers.
Response
Thank you for your valuable suggestion. According to your suggestion, we have unified the formula in the Abstract and Method sections described as below.
Abstract (Line: 20-24)
“The change in SMI (kg/m2/year) was calculated as follows: (SMI at baseline [kg/m2] – SMI at follow-up [kg/m2]) ÷ follow-up period (year). The rate of SMI reduction (%) was calculated as (the change in SMI [kg/m2/year] ÷ SMI at baseline [kg/m2]) × 100. The rate of SMI reduction ≥1.2% was considered as muscle mass loss.”
Method (Line: 154-158)
“The change in SMI (kg/m2/year) was calculated as follows: (SMI at baseline [kg/m2] – SMI at follow-up [kg/m2]) ÷ follow-up period (year). The rate of SMI reduction (%) was calculated as (the change in SMI [kg/m2/year] ÷ SMI at baseline [kg/m2]) × 100 [10]. The rate of SMI reduction ≥1.2% was considered as loss of muscle mass [32].”
Furthermore, we revised the sentence beginning with a number. According to your comment, we have revised it in the Abstract section as below.
Abstract (Line: 24-26)
“Among 197 people with T2DM, 47.2% of them experienced muscle mass loss at 13.7±5.2 months’ follow-up.”
Point 4
The main study variables should be mentioned in the Methods.
Response
Thank you for your suggestion. According to your suggestion, we have mentioned the main study variables mentioned in the Abstract section as below.
Abstract (Line: 28-33)
“Vitamin D intake was related to the incident muscle mass loss after adjusting for age, sex, alcohol consumption, smoking habit, exercise habit, body mass index, SMI, use of glucagon-like peptide-1 agonist, use of sodium glucose cotransporter-2 inhibitor, insulin treatment, glycosylated hemoglobin, creatinine, energy intake, and protein intake (adjusted odds ratio 0.93, 95% confidence interval: 0.88–0.97, p = 0.003).”
Point 5
Conclusions should clearly respond to the previously defined objectives.
Response
Thank you for your comment. According to your comment, we have revised the conclusions responding to the previously defined objectives in the Abstract section as below.
Abstract (Line: 34-37)
“This study revealed that vitamin D intake was related to incident muscle mass loss in older people with T2DM. Vitamin B12 intake tended to be related to incident muscle mass loss, although vitamin A, vitamin B6, vitamin C, and vitamin E intake were not related.”
Point 6
INTRODUCTION
The rationale should be further developed. The authors establish an excessively simplistic relationship between vitamin intake and muscle mass reduction.
Please, the revise the use of the verb ‘to object’ (nothing to do with an objective). Try to be more specific when stating the objective: ‘This prospective cohort study objected to investigate the relationships between the intakes of various vitamins and muscle mass loss in older people with T2DM’: what vitamins? If the objective was only this one, why do you register other nutrients?
Response
Thank you for your comment. The relationship between some vitamin intakes, such as vitamin A, vitamin E, and vitamin B6, and muscle mass loss were not established, although vitamin D intake has been known to relate muscle mass loss in general populations. According to your comments, we have mentioned about vitamin A, vitamin E, vitamin B6, and vitamin C in the Introduction section as below.
Introduction (Line: 66-71)
“In addition, the relationship between some vitamin intakes, such as vitamin A, vitamin B6, and vitamin E, and loss of muscle mass were not established, although vitamin A and vitamin B6 have the effect of activation of immune function [19,20], and vitamin C and vitamin E have the effect of antioxidant [21,22].”
References
- Villamor, E.; Fawzi, W. W. Effects of vitamin a supplementation on immune responses and correlation with clinical outcomes. Clin Microbiol Rev. 2005, 18, 446-464.
- Qian, B.; Shen, S.; Zhang, J.; Jing, P. Effects of Vitamin B6 Deficiency on the Composition and Functional Potential of T Cell Populations. J Immunol Res. 2017, 2017, 2197975.
- Padayatty, S. J.; Katz, A.; Wang, Y.; Eck, P.; Kwon, O.; Lee, J. H.; Chen, S.; Corpe, C.; Dutta, A.; Dutta, S. K.; et al. Vitamin C as an antioxidant: evaluation of its role in disease prevention. J Am Coll Nutr. 2003, 22, 18-35.
- AÄdarkhanov, B. B.; Lokshina, E. A.; Lenskaia, E. G. Molekuliarnye aspekty makhanizma antiokislitel'noÄ aktivnost' vitamina E: osobennosti deÄstviia alpha- i gamma-tokoferolov [Molecular aspects of the mechanism of anti-oxidant activity of vitamin E: features of the action of alpha- and gamma-tocopherol]. Vopr Med Khim. 1989, 35, 2-9.
Furthermore, we have revised the use of the verb ‘to object’, and the objective of the present study were added about each vitamin intakes. According to your comment, we have revised in the Introduction section as below.
Introduction (Line: 71-73)
“The purpose of this prospective cohort study was to investigate the relationships between the intakes of various vitamins and loss of muscle mass in older people with T2DM.”
Moreover, previous studies reported that the association between macronutrient intake, including total energy and protein intake, and incident muscle mass loss. Thus, we have registered other nutrients in this study.
Point 7
METHODS
I strongly encourage authors to follow the STROBE recommendations to report their results.
The subsection ‘Study participants’ should refer only to participants (inclusion and exclusion criteria). ‘Study design’ and ‘Ethics’ should be reported separately.
Response
Thank you for your valuable suggestion. According to your suggestion, we have revised our results to follow the STROBE, and the subsections ‘Study participants’, ‘Study design’ and ‘Ethics’ were reported separately in the Method section described as below.
Method (Line: 75-184)
2.1. Study design
Study design of this study was the prospective cohort study.
2.2. Setting
The KAMOGAWA-DM cohort study, which is an ongoing cohort study of people with diabetes, was introduced in 2014 [23]. This cohort study includes outpatients of Kyoto Prefectural University of Medicine Hospital (Department of Endocrinology and Metabolism, Kyoto, Japan), and Kameoka Municipal Hospital (Department of Diabetology, Kameoka, Japan).
2.3. Study participants
In the present study, we included participants with T2DM who answered to the brief-type self-administered diet history questionnaire (BDHQ) between January 2016 and April 2018 [24]. The exclusion criteria were 1) age under 65 years, 2) missing body composition data at baseline, 3) follow-up duration under 6 months, 4) missing follow-up body composition data, 5) inaccurate data, and 6) extremely low or high energy intake (600 or 4000 kcal/day, respectively) because of low reliability [25].
2.4. Ethics
Written informed consent was obtained from the participants, and the medical data of the participants was compiled into a database with personal information anonymized. The present study was approved by the Research Ethics Committee of Kyoto Prefectural University of Medicine (No. RBMR-E-466-6) and was conducted in accordance with the principles of the Declaration of Helsinki.
2.5. Variables
The present study collected the following data.
Age, sex, family history of diabetes, duration of diabetes, and usage medication data, including sodium glucose cotransporter-2 (SGLT-2) inhibitors, biguanides, glucagon-like peptide-1 (GLP-1) agonist, insulin, and steroids, were collected using medical records.
Lifestyle factors were assessed by a standardized questionnaire. Based on their responses to the questionnaire, participants were grouped into non-smoker and current smoker groups. Moreover, participants were grouped into non-exercisers and exercisers based on their performance or non-performance of any kind of sport at least once a week [24].
Laboratory parameters, including high-density lipoprotein cholesterol, triglycerides, creatinine, fasting plasma glucose, and glycosylated hemoglobin were measured in venous blood samples after overnight fasting. Glycosylated hemoglobin (HbA1c) was measured by high-performance liquid chromatography. The estimated glomerular filtration rate (eGFR; mL/min/1.73 m2) was estimated by follows’ follows; eGFR = 194 × age−0.287 × serum creatinine−1.094 [×0.739 for women] [26].
A multifrequency impedance body composition analyzer, InBody 720 (InBody Japan, Tokyo, Japan), was used for evaluataing body composition. InBody 720 was reported to be good corelation with dual-energy X-ray absorptiometry [27]. Body mass index (BMI, kg/m2), dividing body weight (BW, kg) by height squared (m2), and skeletal muscle mass index (SMI, kg/m2), dividing appendicular muscle mass (kg) by height squared (m2), were calculated. Ideal body weight (IBW) was defined as follows: 22 × participant height squared (m2) [28].
BDHQ was used for habitual food and nutrient intake [25], which is a dietary record to estimate the dietary intake of 58 items of the resondent during the past month. The validity of BDHQ were reported previously [29]. Data of habitual food and nutrient intake (energy [kcal/day]; fat [g/day]; carbohydrate [g/day]; protein [g/day], including vegetable and animal proteins; fiber [g/day]; vitamin A [µg RAE/day]; vitamin B6 [mg/day]; vitamin B12 [µg/day]; vitamin C [mg/day]; vitamin D [µg/day]; vitamin E [mg/day]; and alcohol consumption [g/day], and supplement usage). Energy (kcal/IBW/day), fat (g/IBW/day), and carbohydrate (g/IBW/day) intakes were calculated as energy (kcal/day), fat (g/day) and carbohydrate (g/day) intakes divided by IBW (kg), respectively. Protein (g/BW/day), vegetable protein (g/BW/day) and animal protein (g/BW/day) intakes were calculated as protein (g/day), vegetable protein (g/day) and animal protein (g/day) intakes divided by BW (kg). Carbohydrate-to-fiber intake ratio was calculated as follows: dividng carbohydrate intake by fiber intake [30]. Alcohol consumption was also assessed, and alcohol consumption > 20 g/day were regarded as habitual alcohol consumption [24]. Moreover, data on supplement intake frequency were also obtained, and habitual supplement usage was regarded as using supplements regularly. Energy intake ≥30 kcal/IBW/day dedined as adequate energy intake [12], and protein intake ≥1.2 g/BW/day defined as adequate protein intake [10,31].
2.6. Definition of loss of muscle mass
The change in SMI (kg/m2/year) was calculated as follows: (SMI at baseline [kg/m2] – SMI at follow-up [kg/m2]) ÷ follow-up period (year). The rate of SMI reduction (%) was calculated as (the change in SMI [kg/m2/year] ÷ SMI at baseline [kg/m2]) × 100 [10]. The rate of SMI reduction ≥1.2% was considered as loss of muscle mass [32].
2.7. Statistical method
Continuous variables are presented as means ± standard deviation (SD) or categorical variables are presented as numbr. The statistical significance of differences between groups was evaluated using the chi-square test for categorical variables and student's t-test for continuous variables. Logistic regression analysis was performed to calculate the odds ratio (OR) and 95% confidence interval (CI) for the intake effect of each vitamin on incident loss of muscle mass, adjusting for age, sex, alcohol consumption, smoking habit, exercise habit, BMI, SMI, use of an SGLT2 inhibitor [33], use of a GLP-1 agonist [34], insulin treatment, HbA1c, creatinine, energy intake (kcal/IBW/day), and protein intake (g/BW/day).
Moreover, we examined the effects of vitamin intake on incident loss of muscle mass according to following subgroups: exercise habit; sex; adequate energy intake (≥30 kcal/IBW/day) [12]; adequate protein intake (≥1.2 g/BW/day) [10,31]; usage of supplements; and usage of biguanides. Logistic regression analyses were carried out to calculate ORs and 95% CIs adjusted for age, sex, BMI, energy intake, and protein intake.
Furthermore, the rate of SMI change ≥0.5% [11] and 2.0% [35] were used for other cut-off levels of loss of muscle mass.
EZR (Saitama Medical Center, Jichi Medical University, Saitama, Japan) [36] for statistical analyses and GraphPad Prism, version 8.4.2 (GraphPad Software, Inc., La Jolla, CA, USA), for figure creation. Differences with p values <0.05 were considered statistically significant.
References
- Kobayashi, S.; Honda, S.; Murakami, K.; Sasaki, S.; Okubo, H.; Hirota, N.; Notsu, A.; Fukui, M.; Date, C. Both comprehensive and brief self-administered diet history questionnaires satisfactorily rank nutrient intakes in Japanese adults. J Epidemiol. 2012, 22, 151-159.
- von Haehling, S.; Morley, J. E.; Anker, S. D. An overview of sarcopenia: facts and numbers on prevalence and clinical impact. J Cachexia Sarcopenia Muscle. 2010, 1, 129-133.
Point 8
Please, make a clear distinction between the study variables and potential confounders.
The authors divide participants into two groups according to muscle mass reduction, please state this clearly in the Methods. Why using the cutoff of 1.2% for muscle mass reduction?
Response
Thank you for your comment. According to your comment, we have made a clear distinction between the study variables and potential confounders.
In addition, previous studies showed that muscle mass decreases with age, especially in older people, by 0.5% to 2% per year. Thus, we have set the cut-off of 1.2% per year. However, as you say, the cut-off of 1.2% might be seen as arbitrary. Therefore, according to your comment, we have added other cut-off points of 0.5% and 2%, which have been shown as the cut-off of muscle mass decline in previous studies. Using these cut-off values, similar results were obtained. According to your comment, we have added these points in the Method and Results, including new Table 4, sections described as below.
Method (Line: 153-158)
“2.6. Definition of loss of muscle mass
The change in SMI (kg/m2/year) was calculated as follows: (SMI at baseline [kg/m2] – SMI at follow-up [kg/m2]) ÷ follow-up period (year). The rate of SMI reduction (%) was calculated as (the change in SMI [kg/m2/year] ÷ SMI at baseline [kg/m2]) × 100 [10]. The rate of SMI reduction ≥1.2% was considered as loss of muscle mass [32].”
Previous studies showed that muscle mass decreases with age, especially in older persons, by 0.5% to 2% per year. Thus, we have set the cut-off of 1.2%. However, as you say, the cut-off of 1.2% might be seen as arbitrary. Therefore, according to your comment, we have added other cut-off points of 0.5% and 2%, which have been shown as the cut-off of muscle mass decline in previous studies. Using these cut-off values, similar results were obtained. According to your comment, we have added these points in the new Table 4 and Method and Results sections described as below.
Method (Line: 178-179)
“Furthermore, the rate of SMI change ≥0.5% [11] and 2.0% [35] were used for other cut-off levels of loss of muscle mass.”
Results (Line: 273-278)
“Moreover, Table 4 shows the results of using other cut-off levels of loss of muscle mass of the rate of SMI change ≥0.5% and ≥2.0%. Vitamin D intake was associated with incident loss of muscle mass even at different cutoff levels (cut off point; rate of SMI reduction ≥0.5%, adjusted OR 0.94 [95% CI 0.89-0.99], p = 0.020; and cut off point; rate of SMI reduction ≥2.0%, adjusted OR 0.93 [95% CI 0.88-0.98], p = 0.006).”
References
- von Haehling, S.; Morley, J. E.; Anker, S. D. An overview of sarcopenia: facts and numbers on prevalence and clinical impact. J Cachexia Sarcopenia Muscle. 2010, 1, 129-133.
Point 9
Statistical Analysis: I cannot see the difference between lines 146-152 and lines 153-158. The three models described in Table 3, should be also in the text. Remember, that Tables and manuscript should be understood separately.
Response
Thank you for your comment. Lines 146-152 described the statistical analysis about Table 3, whereas lines 153-158 describes about the statistical analysis about Table S1. As you say, the difference was hard to understand. Thus, we have revised the Method section as below.
Method (Line: 171-177)
“Moreover, we examined the effects of vitamin intake on incident loss of muscle mass according to following subgroups: exercise habit; sex; adequate energy intake (≥30 kcal/IBW/day) [12]; adequate protein intake (≥1.2 g/BW/day) [10,31]; usage of supplements; and usage of biguanides. Logistic regression analyses were carried out to calculate ORs and 95% CIs adjusted for age, sex, BMI, energy intake, and protein intake.”
Point 10
RESULTS
Figure 2. Odds ratio of vitamin D intake on incident muscle mass loss according to the groups. Which groups?
Data included in Figure 2 deserve further explanation in the manuscript.
Response
Thank you for your comment. The descriptions of the subgroups used in Figure 2 and Table S1 were difficult to understand, so we have revised them in the Method section as below.
Method (Line: 171-177)
“Moreover, we examined the effects of vitamin intake on incident loss of muscle mass according to following subgroups: exercise habit; sex; adequate energy intake (≥30 kcal/IBW/day) [12]; adequate protein intake (≥1.2 g/BW/day) [10,31]; usage of supplements; and usage of biguanides. Logistic regression analyses were carried out to calculate ORs and 95% CIs adjusted for age, sex, BMI, energy intake, and protein intake.”
In this study, vitamin D intake was associated with incident muscle mass loss in non-smokers or people with inadequate energy intake. In addition, vitamin D intake was associated with incident muscle mass loss in people both with and without adequate protein intake and in people both with and without exercise habit. We have added these findings in the Results section as below. Furthermore, data included in Figure 2 and Table S1 have been deserved in the Results section as below.
Results (Line: 238-258)
“Vitamin D intake was related to incident loss of muscle mass in people both with (adjusted OR 0.92 [95% CI: 0.86-0.98], p = 0.014) and without exercise habits (adjusted OR 0.92 [95% CI: 0.85-0.99], p = 0.029). Vitamin D was related to incident loss of muscle mass in women (adjusted OR 0.87 [95% CI: 0.79-0.95], p = 0.003) and tended to be related to incident loss of muscle mass in men (adjusted OR 0.95 [95 % CI: 0.89-1.01], p = 0.099), although the latter was not statistically significant. Whereas vitamin D intake was related to incident loss of muscle mass in people without smoking habits (adjusted OR 0.41 [95% CI: 0.20-0.84], p = 0.014), it was not related to incident loss of muscle mass in people with smoking habits (adjusted OR 1.28 [95% CI: 0.03-50.23], p = 0.897). In addition, vitamin D was related to incident loss of muscle mass in people without adequate energy intake (adjusted OR 0.89 [95% CI: 0.83-0.97], p = 0.005), whereas this association was absent in people with adequate energy intake (adjusted OR 0.95 [95% CI: 0.90-1.02], p = 0.137). Vitamin D intake was related to incident loss of muscle mass in people both with (adjusted OR 0.93 [95% CI: 0.88-0.99], p = 0.017) and without adequate protein intake (adjusted OR 0.88 [95% CI: 0.79-0.98], p = 0.020). Vitamin D intake was related to incident loss of muscle mass in people both with (adjusted OR 0.88 [95% CI: 0.78-0.99], p = 0.032) and without usage of supplements (adjusted OR 0.93 [95% CI: 0.87-0.98], p = 0.011).”
Results (Line: 262-272)
“Furthermore, vitamin B12 intake was related to incident loss of muscle mass in people without adequate energy intake (adjusted OR 0.82 [95% CI: 0.70-0.95], p = 0.011). Moreover, vitamin B12 was related to incident loss of muscle mass in people with usage of supplements (adjusted OR 0.71 [95% CI: 0.53-0.94], p = 0.016), whereas this association was absent in people without usage of supplements (adjusted OR 0.96 [95% CI: 0.88-1.05], p = 0.368). Vitamin B12 was related to incident loss of muscle mass in older people without usage of biguanides (adjusted OR 0.88 [95% CI: 0.79-0.97], p = 0.013), whereas this association was absent in older people with usage of biguanides (adjusted OR 1.04 [95% CI: 0.90-1.21], p = 0.569) (Table S1).”
Point 11
DISCUSSION
The brief-type self-administered diet history questionnaire (BDHQ) seems to have important limitations. Please, if available, include metric proprieties and validation studies in Methods and include a comment in the Discussion regarding self-reported methods.
Response
Thank you for your comment. As you say, there is a possibility that BDHQ may not evaluate dietary habits exactly, and we have mentioned this point as one of the limitations of this study in the Discussion section described as below. However, the validation of BDHQ was revealed, and we have added the point in the Method section as below.
Discussion (Line: 372-376)
“Second, the data on eating habits used in this study were obtained at baseline; thus, there is a possibility that eating habits changed during the follow-up period. Furthermore, the data on eating habits being self-reported, eating habits might not be evaluated exactly. However, the specifications and validity of BDHQ were reported previously [29].”
Method (Line: 132-133)
“The validity of BDHQ were reported previously [29].”
References
- Kobayashi, S.; Honda, S.; Murakami, K.; Sasaki, S.; Okubo, H.; Hirota, N.; Notsu, A.; Fukui, M.; Date, C. Both comprehensive and brief self-administered diet history questionnaires satisfactorily rank nutrient intakes in Japanese adults. J Epidemiol. 2012, 22, 151-159.
Round 2
Reviewer 1 Report
The authors made significant changes in the direction of the responses to the review. I am still questioning the threshold for muscle mass loss which is not clear. Either the authors clearly mention why they chose this one, which has to be clinically relevant, or they present the results with different thresholds and change the discussion accordingly.
Author Response
Response to Reviewer 1
The authors made significant changes in the direction of the responses to the review. I am still questioning the threshold for muscle mass loss which is not clear. Either the authors clearly mention why they chose this one, which has to be clinically relevant, or they present the results with different thresholds and change the discussion accordingly.
Response
Thank you for your comment. Previous studies showed that muscle mass decreases with age by 0.5% to 1.2% per year. Thus, we have set the cut-off of 1.2%. However, other studies showed that muscle mass decreases with age by 0.5% to 2.0% per year. Thus, we have added other cut-off points of 0.5% and 2.0%. Consequently, vitamin D intake was associated with muscle mass loss, using not only 1.2% but also other cut-off values of 0.5% and 2.0%.
In addition, we have examined the correlation between the rate of SMI reduction, as continuous variable and vitamin D intake, and clarified that the rate of SMI reduction was correlated with vitamin D intake (r = -0.242, p <0.001). According to your comment, we have added these points in the Method and Results, including new Figure 3, sections described as below.
Method (Line: 164-165)
“Evaluation of correlation was assessed by Pearson’s correlation coefficient.”
Results (Line: 289-291)
“The correlation between vitamin D intake and the rate of SMI reduction were investigated in Figure 3. Vitamin D intake was correlated with the rate of SMI reduction (r = -0.242, p <0.001).”
